# Defining and prioritizing modifiable risk factors towards the co-creation of a urinary incontinence self-management intervention for older men: A sequential multimethod study protocol

**Olawunmi Olagundoye**[1]*, **Shelley Ross**[2], **William Gibson**[1], **Adrian Wagg**[1]

**1** Division of Geriatric Medicine, Department of Medicine, Faculty of Medicine & Dentistry, College of Health Sciences, Edmonton, AB, Canada, **2** Department of Family Medicine, Faculty of Medicine & Dentistry, College of Health Sciences, Edmonton, AB, Canada

* olagundo@ualberta.ca

**Data Availability Statement:** Deidentified research data will be made publicly available when the study is completed and published.

## Abstract

Urinary incontinence (UI), characterized by involuntary urine leakage is a chronic, embarrassing and stigmatizing condition that is under-reported and under-treated). UI is under-prioritized and under-researched, particularly in older men (defined here as men 65+), and there have been calls for more targeted research focusing on this specific group. No existing self-management interventions focus on the needs of older men and none incorporate the perspectives of older men into their development. Furthermore, health inequalities and disparities in continence services for men, and a low level of health seeking behavior in men with UI make it crucial to incorporate their perspectives into intervention development to ensure optimal outcomes. The study will identify risk factors for UI that are potentially amenable to self-management in older men, assess their self-efficacy in managing UI, and determine what modifiable risk factors older men feel are pragmatic to include as part of a self-management program. We will conduct and report a sequential multi-method design consisting of a Delphi study among healthcare experts and a survey among older men with UI, according to the Guidance on Conducting and Reporting Delphi Studies (CREDES) Checklist and the Checklist for Reporting Of Survey Studies (CROSS). A geographically dispersed, multidisciplinary group of 30 health care professionals (urologists, geriatricians, family physicians, and nurses) involved in continence care and a representative sample of at least 128 ethnically diverse older men will participate in a Delphi survey and an older men's survey respectively. The healthcare experts will evaluate an evidence-synthesized list of UI risk factors to determine those potentially amenable to self-management. Delphi rounds will be repeated until consensus threshold of 75% is reached. Thereafter, older men recruited via stratified sampling of population subgroups will rate a list of expert-identified potentially modifiable risk factors to indicate which factors they deem practicable and can prioritize. Older men's survey questionnaires will capture information on patients' characteristics (socio-demographics and UI-related items). The Geriatric Self-Efficacy Index for UI

**Funding:** The UHF fund (Muhlenfeld Family Foundation).

**Competing interests:** The authors have declared that no competing interests exist.

(GSE-UI Index) as well as a Likert scale to assess perceived capability and willingness to modify the expert-identified UI modifiable risk factors will be included. Data will be analyzed quantitatively and qualitatively.

## Introduction

Urinary incontinence is common in older adults, impairs quality of life and increases care needs [1]. In men, it is often a hidden condition with adverse physical and psychosocial consequences [2], in part due to a social trend that feminizes urinary incontinence [3]. It is an independent predictor of the need for institutionalization in men and is associated with an increased likelihood of early death among older men in the community and nursing homes [4]. Although the impact of urinary incontinence (UI) on health-related quality of life in older men and women is similar, most funded research has focused on women [5].

UI is a significant socio-economic problem for older adults, costing the U.S government 16.3 billion dollars in 2001, of which a quarter was for men [6]. In Canada, it was estimated at $8.5 billion annually in 2014 [7]. For male Medicare beneficiaries aged 65 and older in the U. S., the UI economic burden is more than $39 million and exceeds $7000 per individual [8]. Men are also less likely than women to seek healthcare for UI, except in the case of worsening symptoms [9].

UI is not part of the normal aging process, as many older adults have been told, and it can be prevented through various strategies [10].

Self-management is a proven, cost-effective prevention and treatment strategy for managing chronic conditions such as UI. It requires individuals to be involved in identifying challenges and solving problems related to their health problems. It is potentially effective across the prevention spectrum, from the primary prevention of diseases to the secondary prevention of complications and symptom control to the tertiary prevention of disability/ loss of independence [11].

Self-management of chronic diseases focuses on identifying and developing patient-centered strategies to cope with the challenges involved [12], with modifiable risk factors being an important consideration.

Modifiable risk factors are those that can be changed. Some modifiable risk factors associated with the risk of developing chronic diseases in general include tobacco use, alcohol consumption, physical inactivity, overweight/obesity, and poor diet [13] which may be amenable to modification in a self-management program. Patient self-management programs teach individual responsibility and provide tools for chronic illness management [14].

Evidence suggests that psychological factors, such as self-efficacy, play a significant role in UI, thus it is crucial to gain a better understanding of the various factors contributing to UI, including psychological factors that underlie it, in order to develop effective treatment strategies [15]. Self-efficacy is a psychological construct defined as confidence or belief in one's ability to perform certain behaviors, such as preventing unwanted urine loss, according to social-learning theory. It plays a crucial role in initiating and maintaining behavior change [16]. Consequently, it is imperative that older men's self-efficacy for self-management of UI be assessed since self-efficacy improves results for a variety of geriatric conditions [15].

It is clear that self-management packages, which generally address uncomplicated Lower Urinary Tract Symptoms (LUTS) in men, vary in terms of their components, recommendations, and outcomes [17–20]. No existing interventions focus on the needs of older men and

none incorporate the perspectives of older men into their development. In a Cochrane review, researchers found insufficient evidence for lifestyle interventions to treat UI. However, they suggested that weight loss should be a research priority because of emerging evidence for its effect on UI [21]. In their work, Brown *et al* undertook a formal consensus process in defining lifestyle modifications that were likely to be effective in the self-management of uncomplicated LUTS and incorporated these recommendations into a self-management intervention [20]. Based on UCLA's partially anonymous Research and Development Appropriateness Method (RAM), a panel of eight members rated 94 interventions, determining that 57 interventions were appropriate to be incorporated into the self-management program. These interventions were contained within the following categories: patient assessment before starting a self-management program, education and reassurance, fluid management, caffeine, alcohol, concurrent medication, types of toileting, bladder re-training, miscellaneous, and implementation of a self-management program. Weight loss and physical activity were not identified as part of self-management interventions [20]. Brown *et al* also noted that some interventions, for example changes in caffeine and alcohol consumption, may have an adverse effect on quality of life, making them unfeasible [20]. Using the co-creation strategy of intervention mapping [22], our research project will extend existing knowledge by involving older men with UI from the beginning to the end of the intervention development process.

We hypothesize that self-management programs specific to older men may be most effective when the underlying evidence is systematically accessed and then subjected to a formal, rigorous process identifying factors that can be modified by self-management, with input from older men about what they find practicable and willing to change [23].

The proposed study is the second part of a larger project aimed at co-creating a self-management intervention based on synthesized evidence from the literature, and from the perspectives of subject matter experts and of older men with UI. A two-tier formal consensus process will be used to reach agreement among a multidisciplinary group of anonymized experts on what constitutes modifiable risk factors. They will vote on an evidence-synthesized list of risk factors [23], then older men will rate the modifiable risk factors indicating those they deem feasible and can prioritize.

## Objectives

The study aims to identify UI risk factors that older men can self-manage, assess older men's self-efficacy for UI, and identify modifiable risk factors they prioritize as being practical to implement.

## Materials and methods

### Study design

A sequential multi-method design consisting of a Delphi study among healthcare experts and a cross-sectional survey among older men with UI will be employed. Delphi study and older men's survey conduct and reporting will follow the Guidance on Conducting and Reporting Delphi Studies (CREDES) Checklist and the Checklist for Reporting Of Survey Studies (CROSS), respectively [24, 25].

A Delphi survey is the most suitable consensus technique for this study since rapid consensus can be achieved among a wide range of geographically dispersed, multidisciplinary group of experts while maintaining anonymity and preventing groupthink, a common problem with other consensus techniques. As a result, a Delphi survey is relatively more cost-effective to administer and analyze than the others [26].

Healthcare professionals (HCP) will vote on an evidence-synthesized list of UI risk factors in a Delphi survey, maintaining anonymity throughout until consensus is reached. To reduce cognitive load on the Delphi panelists, the list of 98 risk factors identified through a scoping review of UI risk factors in older men has been trimmed down to 46 by grouping similar factors as subcategories of a taxonomical categorization of risk factors as behavioural risk factors, physiological risk factors and age-related physiological changes, demographic factors, medical factors/diseases, environmental factors, and genetic factors [27].

Subsequently, a survey will be conducted to determine if older men with UI perceive themselves as capable and willing to modify any of the modifiable risk factors identified by experts. As an explanatory factor, self-efficacy will be assessed and correlated with perceived capability ratings.

## Participants/eligibility criteria and sampling technique

**Delphi survey.** This will involve a geographically diverse multidisciplinary group of 30 experts, geriatricians, urologists, family physicians and nurses involved in providing healthcare services to older adults including UI patients. It is sufficient to have 30 respondents in a Delphi exercise to achieve consensus and provide statistical rigor [28, 29]. A Delphi panel size greater than 30 may not improve the quality of the results [30]. The next round will not be launched if the response rate of the previous round falls below 70% [28].

Snowball sampling including formal calls for panelists through professional associations such as the International Continence Society (ICS), the Wound, Ostomy and Continence Nursing Association, and the Alberta Primary Care Research Network will be used to assemble the Delphi panel. Experts will also be identified by literature reviews.

Based on the distribution of continence experts in the ICS, the majority (80%) of the Delphi panelists will be urologists, followed by continence nurses (10%) and geriatricians/family physicians (10%).

**Older men's survey.** The survey will involve consenting community-dwelling older men with UI aged 65+.

A sample recruited across ethnic groups will ensure equity, diversity and inclusion by taking into cognizance cultural, language, educational background and socio-economic status differences and ensuring a diverse sample of older men reflective of different population groups.

Gender-based analysis: Based on the World Population Review report that 0.3% of American adults 65 and older are transgender, and Statistics Canada's 2022 census data of 0.04% of older men, gender is unlikely to play a significant role in this study. Considering the likely sample size, any conclusions derived from such an analysis would be severely limited in validity.

## Sample size estimation and sampling technique

At 95% confidence level, desired precision of 5% and a prevalence value of 0.092 for UI in men 65 years and older from a Canadian Community Health Survey [31], a minimum sample size of 128 was derived. The sample size will be distributed based on the sizes of demographic strata using stratified sampling of older men across ethnic strata. According to the Canada 2021 census, a minimum of 84 (65.4%) Caucasians and 36 (27.8%) visible minority/racialized group (including South Asian, Filipino, Black, and Chinese) will be recruited [32].

## Data collection procedure and instruments

**Delphi study.** For the Delphi survey, an expert on the research team will frame the instructions for participants and develop the questionnaire using clear and precise language. A

coordinator will organize, coordinate the Delphi process, record and analyze results from each round of Delphi, and write the reports. An online survey link to a consent form and semi-structured questionnaire will be emailed to the experts individually. It will be accompanied by a letter of transmittal that describes the purpose of the Delphi and states the anticipated time-lines for each round. The questionnaire will itemize the risk factors, identified through a scoping of review of risk factors for older men [33], to be voted on, and provide a space for comments and justifications for choices (S1 Appendix). They will indicate whether each factor is important for the development of UI and whether older men can modify them. Feedback will be provided to the experts after each round, to allow them to review their forecasts or change their opinion on the basis of the group consensus [26]. Rounds will be repeated until consensus threshold of 75% is reached [34]. The Delphi workflow is shown in Fig 1.

**Older men's survey.**   For the rating survey among older men with UI, an eligibility survey testing for the inclusion criteria (older men age 65+ and history of UI using the UI severity index by Sandvik et al [35]) will precede the main survey (S2 Appendix). The main survey will include a participant information sheet and consent form, a section for participant characteristics including UI-related items, another for the Geriatric Self-Efficacy Index for UI (GSE-UI Index) and one for two sets of a 5-point Likert rating scale presented at a Grade 8 reading level. In the first set, assessing willingness to change, a point will represent "I DO NOT want to make a change, I might be willing to make a change, I am pretty sure that I want to make a change, I definitely want to make a change, and Does not apply to me" In the second set assessing perceived capability to change, a point will stand for "I am NOT able to make a change, I might be able to make a change, I am pretty sure that I am able make a change, I am definitely able to make a change, and Does not apply to me." in response to two different instructions on the Likert scale "(1) For the things on the list, please choose the answer that is closest to how you feel about your willingness to make a change. (2) For the things on the list, please choose the answer that is closest to how you feel about your ability to make a change." In the participant characteristics and clinical information section, UI-related quality of life will be evaluated with the Incontinence Quality of Life (I-QOL) instrument [36, 37], subjective socioeconomic status will be determined using a 5-point scale to address an item on perceived financial situation/socioeconomic status [38] and items assessing UI frequency, severity, type and treatment. They will also be asked to indicate which mode of intervention delivery they prefer.

Depending on the participants' preference, questionnaires will be self-administered or interviewer-administered. To minimize recruitment bias, self-administered questionnaires will be electronic or paper-based. Depending on the preferences of potential participants, a copy of the questionnaire will be sent by email or postal mail.

The UI Severity Index developed by Sandvik et al has been recommended for routine use as a valid, reliable, short, simple, and sensitive measure of UI. It showed a good test-retest reliability ($\kappa = 0.78$, $P < 0.001$) and validity ($r = 0.55$, $P < 0.001$) [35, 39].

The Incontinence Quality of Life (I-QOL) instrument is a validated self-report measure of quality of life specific to urinary incontinence, developed by Wagner *et al*. for research and patient care. It has high internal consistency with Cronbach's alpha of 0.95 and high test-retest reliability of 0.93 [36].

The Geriatric Self-Efficacy Index for UI (GSE-UI Index): A 20-item questionnaire developed by Tannenbaum *et al*., published in 2008 [15], and later modified to 12 items [40]. It allows measurement of a person's confidence in their ability to prevent urine loss. It has high internal consistency with a Cronbach alpha of 0.90 and good to excellent item test-retest reliability with intra-class correlation coefficients (ICC) greater than 0.6 [40].

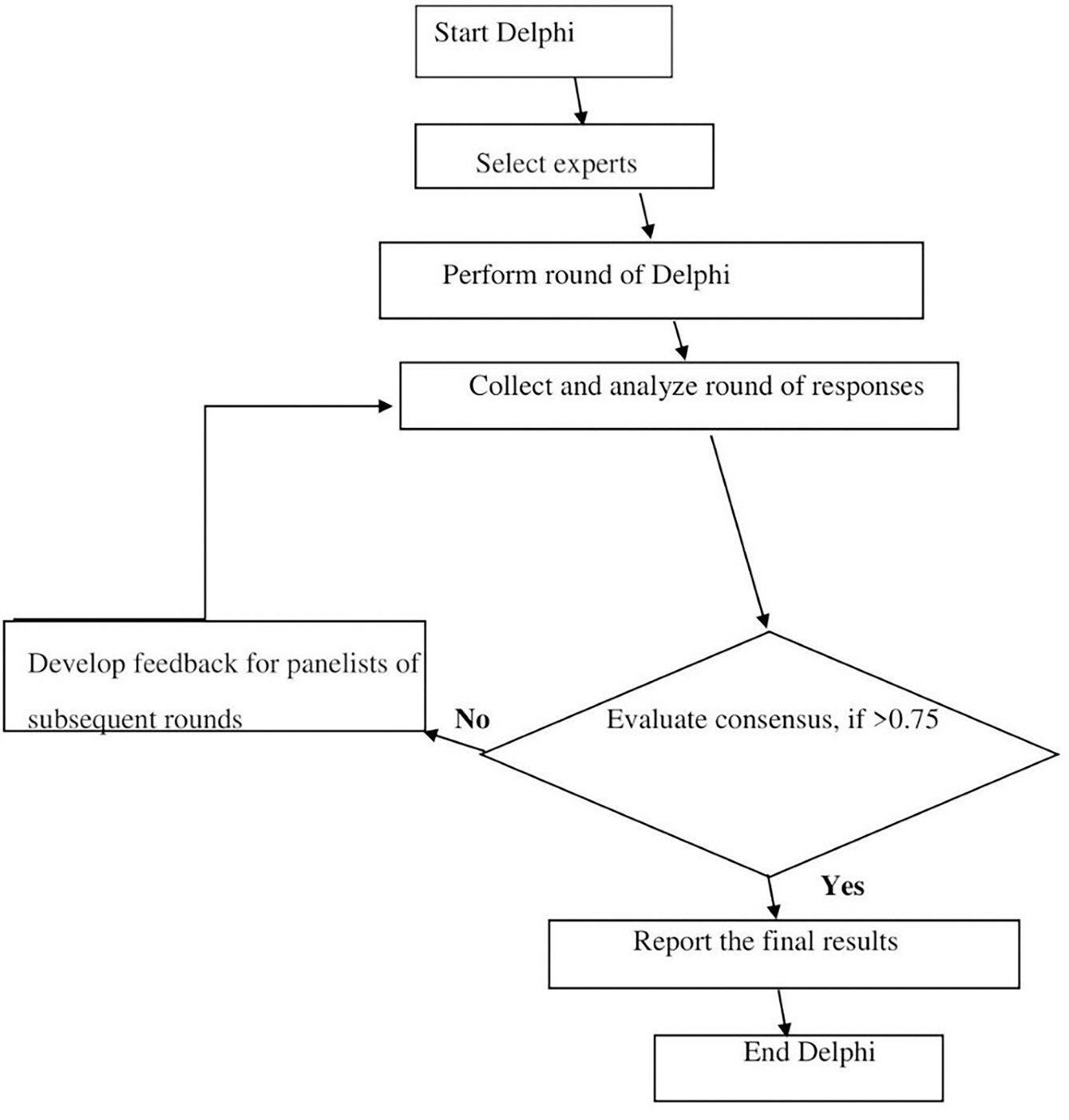

**Fig 1. Delphi workflow.**

## Pilot testing

Older men's questionnaires will be pilot-tested for face validity and reliability (internal consistency) among a pilot sample of 30 older men who would not participate in the main study.

## Survey administration

Recruitment of older men with UI will be conducted through seniors' associations, ethnic groups, retirement living facilities, religious organizations and Alberta Health Services (AHS) facilities.

We will distribute infographics embedded with QR codes linking to screening/eligibility questions to associations/groups after obtaining permissions from management and consent from participants. REDCap survey tool will be used, and survey data will be protected.

Six months will be the timeframe for the survey. Data entry errors for paper-based questionnaires will be mitigated by double checking inputted data against each questionnaire. To prevent multiple participations by online survey participants, links to the survey will be emailed to those who meet the eligibility criteria (age 65+ and history of accidental urine leakage).

## Statistical analysis

To reduce non-response bias, the sampling weights of respondents will be increased through a weighting adjustment to compensate for the non-respondents.

Results will be analyzed for agreement in each round until consensus is reached. The justifications for a Yes/No vote will be analyzed using qualitative content analysis, and will be incorporated into the next round of questions in an anonymized format.

Data will be analyzed using the Statistical Package for Social Sciences version 23.0 (IBM Corp.2015, Armonk, NY). Continuous variables will be presented in means and standard deviation (SD), while categorical variables will be presented in proportions.

## Ethical clearance

This study protocol has been approved by the University of Alberta's Health Research and Ethics Board (Ethics ID: Pro00134947). Permissions will be obtained from the various associations, operational approval will be obtained from the Alberta Health Services facilities, and the research will be conducted with informed consent from all participants.

# Results

Following the CREDES checklist, we will present Delphi results for each round to reflect the evolution of consensus over the rounds transparently, with figures highlighting the average group response and changes between rounds. Modifications to the survey instrument based on previous rounds will also be reported [24].

Similarly, older men's survey results will be presented in texts, tables and charts. Participants' characteristics and assessed outcomes will be described.

# Supporting information

**S1 Appendix. Delphi survey.**
(DOCX)

**S2 Appendix. Older men's survey questionnaire.**
(DOCX)

## Acknowledgments

As a recipient of the Dr. Peter N. McCracken Legacy Scholarship, OO gratefully acknowledges the support of the Glenrose Rehabilitation Foundation and Alberta Health Services. The research study will contribute towards a PhD degree award for OO.

## Author Contributions

**Conceptualization:** Olawunmi Olagundoye, Adrian Wagg.

**Funding acquisition:** Adrian Wagg.

**Investigation:** William Gibson.

**Methodology:** Olawunmi Olagundoye, William Gibson, Adrian Wagg.

**Project administration:** Olawunmi Olagundoye, Adrian Wagg.

**Resources:** Adrian Wagg.

**Supervision:** Adrian Wagg.

**Validation:** Shelley Ross.

**Writing – original draft:** Olawunmi Olagundoye.

**Writing – review & editing:** Olawunmi Olagundoye, Shelley Ross, William Gibson, Adrian Wagg.

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
