## [Decision Letter · Decision Letter 0]

23 May 2024

Defining and prioritizing modifiable risk factors towards the co-creation of a urinary incontinence self-management intervention for older men: A sequential multimethod study protocol

PONE-D-24-03156

Dear Dr. Olawunmi Olagundoye,

We’re pleased to inform you that your manuscript has been judged scientifically suitable for publication and will be formally accepted for publication once it meets all outstanding technical requirements.

Kind regards,

Malgorzata Wojcik, Ph.D

Academic Editor

PLOS ONE

Journal Requirements:

"The UHF fund (Muhlenfeld Family Foundation)"

Please respond by return e-mail so that we can amend your financial disclosure and competing interests on your behalf.

Additional Editor Comments (optional):

Reviewers' comments:

Reviewer's Responses to Questions

**Comments to the Author**

1. Does the manuscript provide a valid rationale for the proposed study, with clearly identified and justified research questions?

Reviewer #1: Yes

2. Is the protocol technically sound and planned in a manner that will lead to a meaningful outcome and allow testing the stated hypotheses?

Reviewer #1: Yes

3. Is the methodology feasible and described in sufficient detail to allow the work to be replicable?

Reviewer #1: Yes

4. Have the authors described where all data underlying the findings will be made available when the study is complete?

Reviewer #1: Yes

5. Is the manuscript presented in an intelligible fashion and written in standard English?

Reviewer #1: Yes

6. Review Comments to the Author

You may also provide optional suggestions and comments to authors that they might find helpful in planning their study.

Reviewer #1: The article was written in a clear and transparent manner. The protocol presented here is easily re-performed by other authors who would like to conduct similar research

7. PLOS authors have the option to publish the peer review history of their article (what does this mean?). If published, this will include your full peer review and any attached files.

Reviewer #1: No

---

## [Editor Report · Acceptance letter]

20 Jun 2024

PONE-D-24-03156 

PLOS ONE

Dear Dr. Olagundoye, 

I'm pleased to inform you that your manuscript has been deemed suitable for publication in PLOS ONE. Congratulations! Your manuscript is now being handed over to our production team.

Kind regards, 

on behalf of

Dr. Malgorzata Wojcik 

Academic Editor

PLOS ONE